# Bio-Augmentation of $S^{2-}$ Oxidation for a Heavily Polluted River by a Mixed Culture Microbial Consortium

Chen Song [1,2], Yajun Shi [1], Hongjie Gao [1], Ping Liu [3] and Xiaoling Liu [1,*]

[1] State Key Laboratory of Environmental Criteria and Risk Assessment, Chinese Research Academy of Environmental Sciences, Beijing 100012, China; s--c@163.com (C.S.); marie.shiyj@hotmail.com (Y.S.); gaohj@craes.org.cn (H.G.)

[2] Nanjing Lotus High-Tech Material Technology Co., Ltd., Nanjing 210029, China

[3] Technical Center for Soil, Agricultural and Rural Ecology and Environment, Ministry of Ecology and Environment, Beijing 100012, China; liuping@tcare-mee.cn

* Correspondence: liuxl@craes.org.cn; Tel.: +86-15011019493

**Abstract:** The redox balance of inorganic sulfur in heavily polluted rivers might be disrupted, making sulfur reduction a major metabolic pathway of sulfate-reducing bacteria (SRB), leading to a massive accumulation of $S^{2-}$ and blackening the water bodies. A mixed culture microbial consortium (MCMC) of *Citrobacter* sp.sp1, *Ochrobactrum* sp.sp2, and *Stenotrophomonas* sp.sp3 was used to activate native sulfate-oxidizing bacteria (SOB) to augment the $S^{2-}$ oxidizing process. The results demonstrated that MCMC had a significant sulfur oxidation effect, with 98% $S^{2-}$ removal efficiency within 50 h. The sulfide species varied greatly and were all finally oxidized to $SO_4^{2-}$. The mechanism of bio-augmentation was revealed through high throughput sequencing analysis. The MCMC could stimulate and simplify the community structure to cope with the sulfide change. The microorganisms (family level) including *Enterococcaceae*, *Flavobacteriaceae*, *Comamonadaceae*, *Methylophilaceae*, *Caulobacteraceae*, *Rhodobacteraceae*, and *Burkholderiaceae* were thought to be associated with sulfide metabolism through the significant microbial abundance difference in the bio-treatment group and control group. Further Pearson correlation analysis inferred the functions of different microorganisms: *Comamonadaceae*, *Burkholderiaceae*, *Alcaligenaceae*, *Methylophilaceae*, and *Caulobacteraceae* played important roles in $S^{2-}$ oxidization and $SO_4^{2-}$ accumulation; and *Comamonadaceae*, *Burkholderiaceae*, *Alcaligenaceae*, *Methylophilaceae*, *Caulobacteraceae*, *Campylobacteraceae*, *Bacteriovoracaceae*, and *Rhodobacteraceae* promoted the sulfur oxidation during the whole process.

**Keywords:** bio-augmentation; $S^{2-}$ oxidation; mixed culture; SOB; heavily polluted rivers





## 1. Introduction

In recent decades, much $S^{2-}$ has been released into the rivers with the intensification of industry and anthropogenic activity, resulting in the accumulation of inorganic sulfur in natural water, especially in heavily polluted rivers [1]. Studies have found that the accumulation of $S^{2-}$ is one of the main causes of river black odor in China [2]. The black-odor phenomenon in water bodies not only leads to the destruction of the original ecological function but also seriously affects the regular life of the surrounding residents [3]. The blackened river is mainly formed by the combination of $S^{2-}$ and metal ions in the water bodies, which affects the survival and reproduction of plants and animals in the water body and leads to abnormal microbial activity [2,3]. This phenomenon has become one of the prominent water environment problems in China. Song et al. found that the $S^{2-}$ concentrations ranged from 3.84 to 17.93 mg/L along Dongsha River in Beijing, which was a heavily polluted river [4]. Sheng et al. also found the mean concentration of $S^{2-}$ could reach 3.15 mg/L in the Dihe River in China [5]. The standard of sulfide from the Ministry of Environment and Protection of the People's Republic of China GB 3838-2002 (MEP) divided the water qualities into five levels, according to the environmental functions

and protection objectives of water bodiesand the maximum concentration standard (level V) for sulfide is $\leq 1.0$ mg/L [6]. The high concentration of $S^{2-}$ in a heavily polluted river largely contributes to the generation and accumulation of hydrogen sulfide and insoluble metal sulfides due to the combination of $S^{2-}$ and hydrogen or heavy metal ions in water bodies [7]. Hydrogen sulfide, a biologically important molecule with complex physiological functions, can cause potential inhalation hazards as well as an inhibitory effect on cellular respiration [8]. While metal sulfides such as CuS, MnS, and FeS are often suspended in the river, and further aggravate the river blackening [2]. $S^{2-}$ in a river comes from two pathways: one is exogenous input, and the other is endogenous release [4]. Exogenous input mainly refers to the discharge of $S^{2-}$-containing wastewater from industry, agriculture, vehicular traffic, and other sources [9]. Endogenous release includes three major aspects: (1) $S^{2-}$ liberation directly from river sediment due to the scouring effect [10]; (2) the biodegradation of sulfur-containing high molecular substances such as proteins under the action of native microorganisms [11]; and (3) the bio-reduction of oxidized sulfides as a result of the sulfate-reducing bacteria (SRB) effect [12] (Sorokin et al. 2014).

Inorganic sulfur bio-cycling is an important part of material circulation in nature. In general, $S^{2-}$ is bio-oxidized to high-valence sulfur comprising $S^0$, $S_2O_3^{2-}$ as well as $SO_3^{2-}$, and all of them are finally bio-oxidized to $SO_4^{2-}$ [13,14]. Conversely, $SO_4^{2-}$ can be bio-reduced to $SO_3^{2-}$, $S_2O_3^{2-}$, and $S^0$, and these low-valence sulfurs are further bio-reduced to $S^{2-}$. Sulfate-oxidizing bacteria (SOB) are responsible for the bio-oxidation of sulfur, whereas SRB is in charge of the bio-reduction of sulfur [12]. These two paths, as important bio-geochemical processes in nature, maintain the balance of $S^{2-}$ redox and enable its concentration to remain at a low level in native rivers [15,16]. Some studies showed that in a heavily polluted river, dissolved oxygen was often depleted, and the ability of water bodies' re-oxygenation was greatly weakened, which led to the metabolic block in the bio-oxidation pathway of sulfur [17]. On the other hand, the activities of SRB are significantly enhanced in an oxygen-free environment, which accelerates the bio-reduction process of sulfur, resulting in the large accumulation of $S^{2-}$ and the further exacerbation of river quality [2]. Inoculating special strains, especially mixed culture, has been proven to be a good method to remove pollutants due to the microbial diversity or abundant enzyme system [12,18,19]. Moreover, SOB obtained from the black-odor river could better adapt to the conditions of the original black-odor river, have better survivability in treating the river, and have higher processing efficiency than the SOB obtained from other sources. Therefore, using SOB with mixed culture screened from a heavily polluted river may be an effective measure to eliminate $S^{2-}$ pollution. SOB is a large topic that has attracted attention around the world. To date, many pure SOB have been isolated mainly from sludge [19,20], and these SOB are widely applied in bioleaching and chemical wastewater treatment [21,22]. Gevertz et al. isolated SOB from an oil field in Canada to reduce/cycle the sulfide and elemental sulfur [23]; Brock et al. isolated *Sulfolobus* as a potential geochemical agent in high-temperature hydrothermal systems [24]; Anandham et al. isolated SOB from rhizosphere soils and discussed the possible thiosulfate oxidation pathways of the SOB and the phylogenetic distribution of the sulfur oxidation gene (*soxB*) [25]. In comparison, less work has been carried out with the SOB under the condition of mixed culture isolated from a river; the mechanism of $S^{2-}$ removal involved is also less reported.

In this study, three strains with better sulfur oxidation effects were isolated from the Dongsha River, a heavily polluted river located in Beijing, China. These strains, including *Citrobacter* sp.sp1, *Ochrobactrum* sp.sp2, and *Stenotrophomonas* sp.sp3, were used to remove $S^{2-}$ with mixed culture in our previous work [4,26]. Although the consortium in previous work showed the potential to use $S^{2-}$ as electron donors, the mechanism involved in the $S^{2-}$ oxidation process was still unclear. Therefore, the purpose of this article was to investigate the bio-augmentation process of $S^{2-}$ oxidizing and increase the understanding of the involved bio-augmentation mechanism. In order to achieve this purpose, we added a mixed culture microbial consortium (MCMC) into a bio-treatment group and set up a

control group (without MCMC) to analyze the effect of the MCMC on the structure of the original microbial community using diversity analysis; infer the sulfate metabolism bacterium (SMB) that might be activated by the MCMC through difference analysis of the bio-treatment group and the control group; explore the mechanism by high throughput sequencing analysis and the correlation analysis between sulfide and SMB; and infer the key role of these microorganisms in each of the sulfur oxidation process. These results will help provide fundamental information for the treatment of heavily polluted rivers in the future and understand the bio-augmentation effect on $S^{2-}$ oxidation.

## 2. Materials and Methods

### 2.1. Water Sample

The water sample was collected from the Dongsha River. The characteristics of the water sample were analyzed, and the concentrations of $S^{2-}$, chemical oxygen demand (COD), ammonia (NH$_3$-H), and total phosphorus (TP) were $20.7 \pm 1.2$ mg/L, $104.5 \pm 8.0$ mg/L, $5.3 \pm 0.4$ mg/L, and $3.0 \pm 0.1$ mg/L, respectively. The water samples were collected in September, the temperature at the time was 25 °C; the water body of the Dongsha River was stagnant, not flowing, with some algae on the surface and no fish in it.

### 2.2. The Strains

According to our previous study [4,26], *Citrobacter* sp. sp1, *Ochrobactrum* sp. sp2, and *Stenotrophomonas* sp. sp3 were all isolated from the Dongsha River of Beijing, which was heavily polluted [4], and each of them had been proved to oxidize $S^{2-}$ efficiently. The sequences of these three strains were all uploaded to The National Center for Biotechnology Information (NCBI), and their GenBank accession numbers were MH181794, MH181795, and MH181796, respectively. They had also been deposited in China General Microbiological Culture Collection Center (CGMCC), and the strain numbers were CGMCC No. 24337, 24338, and 18394, respectively.

### 2.3. The MCMC

The MCMC consisted of *Citrobacter* sp. sp1, *Ochrobactrum* sp. sp2, and *Stenotrophomonas* sp. sp3 in a proportion of 1:1:1, and the initial cell density of each strain was $8.96 \times 10^5$ cells/mL (Colony-Forming Units, Cfu). The MCMC was cultured using a liquid medium containing 20 g/L of glucose, 10 g/L of peptone, and 10 g/L of yeast extract with a pH of 7. All experiments were carried out in 250 mL conical flasks containing 150 mL sterilized liquid medium, and the cultivations were conducted on a rotary shaker with 120 rpm at 25 °C. After 48 h, the MCMC was harvested for the following sulfur oxidation experiments.

### 2.4. Experiment of Sulfur Oxidation

The experiment was divided into two groups. One was the bio-treatment group that was inoculated with 0.1 g/L of the MCMC, while the other was the control group without the MCMC. The sulfur oxidation process was investigated using the water sample from the Dongsha River in a conical flask with 800 mL of working volume. All of the experiments were run on a rotary shaker at 25 °C and 120 rpm. Regularly, samples were taken out for analysis. All experiments were operated in triplicate.

#### 2.4.1. DNA Extraction and PCR Amplification

Microbial DNA was extracted from the samples using the E.Z.N.A.® soil DNA Kit (Omega Bio-tek, Norcross, GA, USA) according to the manufacturer's protocols. The V3-V4 region of the prokaryote 16S ribosomal RNA gene was amplified by the polymerase chain reaction (PCR). The procedure of PCR was as follows: 95 °C for 3 min, followed by 27 cycles at 95 °C for 30 s, 55 °C for 30 s, 72 °C for 45 s, and a final extension at 72 °C for 10 min. The primers for prokaryote were 338F (5′-ACTCCTACGGGAGGCAGCAG-3′) and 806R (5′-GGACTACHVGGGTWTCTAAT-3′), respectively. The barcode was an eight-base sequence unique to each sample. PCR reactions were performed in triplicate in a 20 μL

mixture containing 4 μL of 5 × FastPfu Buffer, 2 μL of 2.5 mM dNTPs, 0.8 μL of each primer (5 μM), 0.4 μL of FastPfu Polymerase, and 10 ng of template DNA.

### 2.4.2. Illumina MiSeq Sequencing

Amplicons were extracted using 2% agarose gels and purified with an AxyPrep DNA Gel Extraction Kit (Axygen Biosciences, Union City, CA, USA) according to the manufacturer's instructions. The purified amplicons were then quantified using a QuantiFluor™-ST Kit (Promega, Madison, WI, USA). The Purified amplicons were pooled in equimolar and paired-end sequenced (2 × 250) on an Illumina MiSeq platform according to the standard protocols. The raw reads were deposited into Sequence Read Archive (SRA) database (Accession Number: PRJNA394809) of the NCBI.

### 2.4.3. Processing of Sequencing Data

Raw fastq files were de-multiplexed and quality-filtered using QIIME (version 1.17) with the following conditions: (1) the 300 bp reads were truncated at any site receiving an average quality score <20 over a 50 bp sliding window, discarding the truncated reads that were shorter than 50 bp; (2) exact barcode matching, 2 nucleotide mismatch in primer matching, reads containing ambiguous characters were removed; (3) only sequences that overlap longer than 10 bp were assembled according to their overlap sequence. Reads that could not be assembled were discarded.

Operational Units (OTUs) were clustered with a 97% similarity cutoff using UPARSE (version 7.1 http://drive5.com/uparse/, accessed on 28 January 2017), and chimeric sequences were identified and removed using UCHIME. The taxonomy of each 16S rRNA gene sequence was analyzed using RDP Classifier (http://rdp.cme.msu.edu/, accessed on 28 January 2017) and then compared using the silva 16S rRNA gene reference database (SSU128). The confidence threshold was set at 70% [27].

### *2.5. Analytical Methods*

The concentrations of $S^{2-}$, $S^0$, $S_2O_3^{2-}$, $SO_3^{2-}$, and $SO_4^{2-}$ were separately analyzed according to Chinese water and wastewater monitoring and analysis methods, respectively [6]. A Uv-vis spectrophotometry (Uv-1800, Kyoto, Japan) was used to determine the concentrations of $S^{2-}$, $S^0$, and $SO_4^{2-}$; while the concentration of $SO_3^{2-}$ was measured using an ion chromatograph (ISC-1100, Waltham, USA). Meanwhile, $S_2O_3^{2-}$ was measured according to the titrimetric method.

The relationship between sulfur oxidization and SMB was analyzed using SPSS statistical software (version 17.0, New York, NY, USA). "*" and "**" meant that the correlation was significant at the 0.05 level and 0.01 level, respectively.

## 3. Results

### *3.1. Variation of Inorganic Sulfur as Result of MCMC*

The concentrations of $S^{2-}$, $S^0$, $S_2O_3^{2-}$, $SO_3^{2-}$, and $SO_4^{2-}$ were detected throughout the whole process, and the results are shown in Figure 1. As shown in Figure 1a, for the sample inoculated with the MCMC, the variations of different chemical forms of sulfur were significant during 50 h of the oxidization process. The concentration of $S^{2-}$ gradually decreased as the bio-treatment time was extended, and its final concentration declined to less than 1.0 mg/L. The main reason for the decreasing concentration of $S^{2-}$ was that $S^{2-}$ was oxidized to generate other chemical forms of sulfur such as $S^0$ with the action of MCMC. This resulted in the rapid accumulation of $S^0$ before the first 18 h and the peak concentration reached 0.4 mg/L. Similarly, $S_2O_3^{2-}$ and $SO_3^{2-}$ appeared at a peak concentration at 18 h and 28 h, respectively. The maximal value of $S_2O_3^{2-}$ arrived at 57.9 mg/L, while for $SO_3^{2-}$ it was 51.7 mg/L. In the subsequent oxidation process, the concentrations of $S^0$, $S_2O_3^{2-}$, and $SO_3^{2-}$ decreased significantly. On the other hand, the concentration of $SO_4^{2-}$ was clearly raised throughout the process, especially at the initial 18 h, and the final concentration was up to 43.0 mg/L. As for the control without the MCMC, a part of sulfur chemical forms

such as $S^{2-}$, $S^0$, and $SO_3^{2-}$ were determined (Figure 1b). The concentration of $S^{2-}$ fell initially, and then slightly raised during the later process, while other sulfur chemical forms kept relatively stable in the whole process.

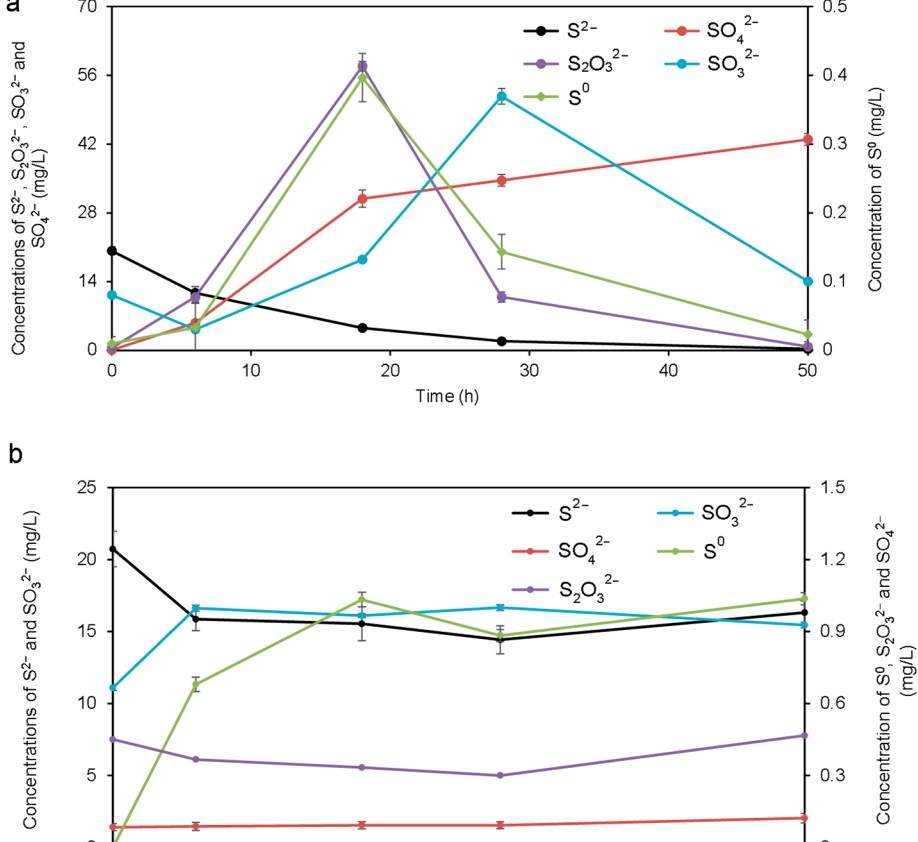

**Figure 1.** Variation of inorganic sulfur during the whole oxidization with the (**a**) bio-treatment group and (**b**) the control group.

### 3.2. Variation of Community Structure as a Result of MCMC

Inoculation with special strains to augment bio-reaction occurs mainly through two processes: (1) the formation of stable dominant microorganisms from inoculation in the whole process; and (2) the variation of community structure due to the simulation of original microorganisms by inoculation [28,29]. A comparison of the microbial diversity between the bio-treatment group and the control group was evaluated, as shown in Table 1. In general, different samples showed a relatively similar trend in terms of richness. Furthermore, as indicated by Ace, Chao, Shannon, and Coverage indices implied that there were similar diversity dynamics of the microbial community in the two groups.

**Table 1.** Comparison of microbial diversity between the bio-treatment group and the control group.

| Item | Bio-Treatment Group | | | | | Control Group | | | | |
|---|---|---|---|---|---|---|---|---|---|---|
| | 1 h | 6 h | 18 h | 28 h | 50 h | 1 h | 6 h | 18 h | 28 h | 50 h |
| Ace | 452.97 | 491.66 | 500.24 | 513.91 | 481.42 | 497.68 | 502.67 | 549.58 | 492.88 | 444.55 |
| Shannon | 2.36 | 2.27 | 3.53 | 4.13 | 4.42 | 3.87 | 3.51 | 3.17 | 4.25 | 4.52 |
| Coverage | 0.997 | 0.997 | 0.998 | 0.998 | 0.997 | 0.999 | 0.998 | 0.997 | 0.998 | 0.998 |

The difference of community in two groups with or without the cultured MCMC was revealed using PCoa as shown in Figure 2. With regard to principal component 1 (PC1), all samples obtained from the control groups clustered on the left side of the figure, while the samples from the bio-treatment groups covered both the left and right sides. For PC2, each prokaryote sample in the bio-treatment groups was more widely dispersed than other samples in the control groups.

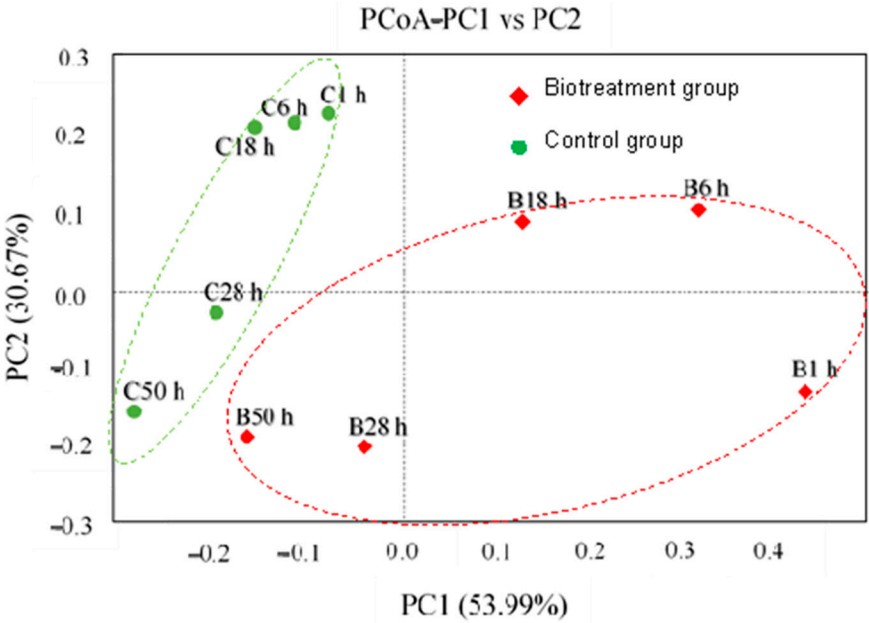

**Figure 2.** PCoa analysis of two groups with or without the MCMC.

As shown in Figure 2, the addition of sulfur-oxidizing microorganisms resulted in a significant difference between the two groups. The first 18 h of the bio-reaction process in the bio-treatment group were significantly altered by the MCMC; but for the control group, there was no obvious difference. As the reaction progressed, the differences between the two groups gradually decreased, and at 50 h, the samples in the group with the MCMC (B50 h) tended to be similar to those in the control group (C50 h).

### 3.3. Analysis of Sulfides Metabolism Bacterium

The differences in microbial abundance between the bio-treatment group and the control group during the reaction were investigated to extrapolate the potential SMB involved in the inorganic sulfides' bio-oxidation process (only species with mean sums in the top 10 showed). After the addition of the MCMC, *Enterococcaceae* only showed up in the bio-treatment group after the reaction started. Throughout the reaction process, the abundance of *Enterococcaceae*, *Flavobacteriaceae*, *Comamonadaceae*, *Methylophilaceae*, *Caulobacteraceae*, *Rhodobacteraceae*, and *Burkholderiaceae* in the bio-treatment group showed significant differences compared with the control group. Most microorganisms play a role in sulfur oxidation, but it cannot be excluded that some microorganisms are associated with sulfur reduction. Moreover, the abundance of *Enterococcaceae*, *Comamonadaceae*, *Flavobacteriaceae*, *Caulobacteraceae*, *Rhodobacteraceae*, *Sphingomonadaceae*, *Burkholderiaceae*, and *Bacteriovoracaceae* varied considerably over the reaction time. This suggested that these microorganisms play different roles in the different valence states of sulfates for sulfur oxidation.

After 6 h of the reaction (Figure 3a), in addition to *Enterococcaceae* and *Flavobacteriaceae*, the abundance of *Caulobacteraceae* also showed significant differences between the two groups, and at this time, only the abundance of *Enterococcaceae* was significantly higher in the bio-treatment group with the MCMC than in the control group. During the first 28 h, the abundance of *Flavobacteriaceae*, *Sphingomonadaceae*, and *Cryomorphaceae* in the control group was consistently higher than the corresponding abundance in the bio-treatment group. As

the reaction continued, during the period of 18–50 h, the abundance of 14 microorganisms, including *Caulobacteraceae*, *Rhodobacteraceae*, and *Sphingomonadaceae*, was found to change obviously with the reaction process in both groups (Figure 3b–d). In addition, taxonomically, *Citrobacter* sp.sp1, *Ochrobactrum* sp.sp2, and *Stenotrophomonas* spsp3. were classified as *Enterobacteriaceae*, *Brucellaceae*, and *Xanthomonadaceae*, respectively (the sum of means outside the top 10 species).

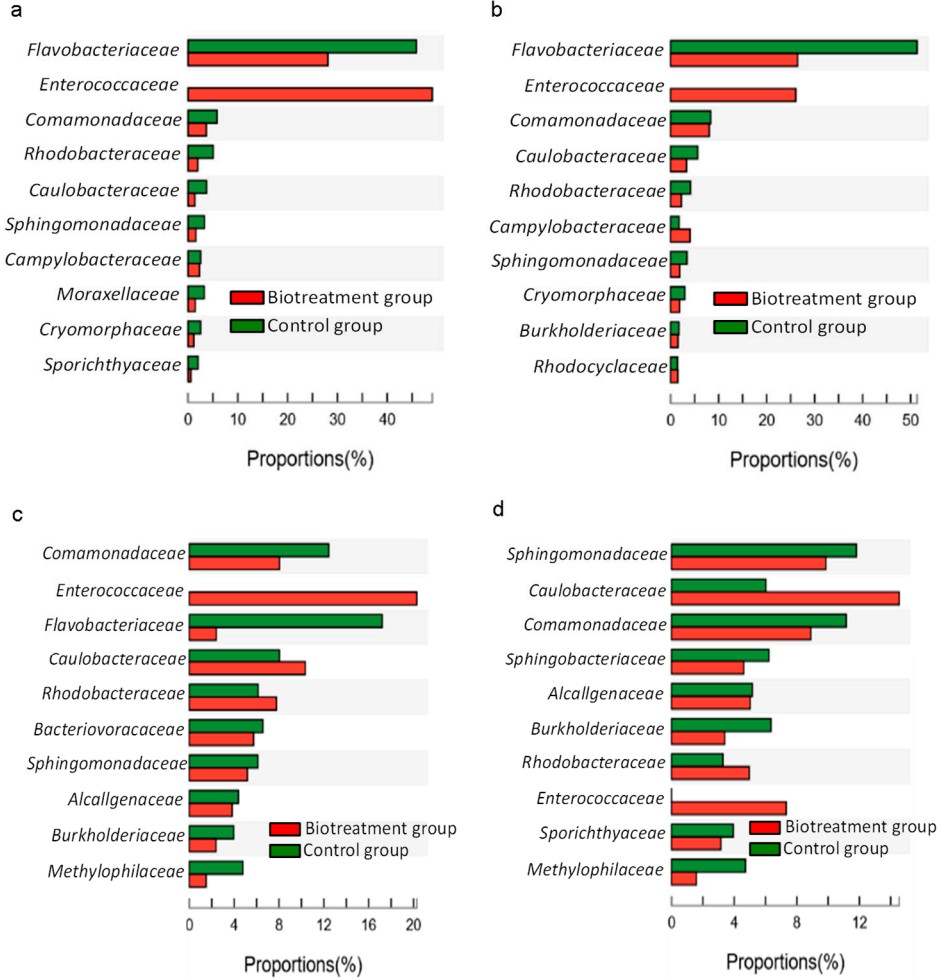

**Figure 3.** Abundance difference between the bio-treatment group and the control group ((**a–d**) Prokaryote difference at 6 h, 18 h, 28 h, and 50 h, respectively).

Differences in $S^{2-}$, $S^0$, $S_2O_3^{2-}$, and $SO_4^{2-}$ concentrations were obvious in the two groups at 18 h, while $SO_3^{2-}$ concentration showed a great difference at 28 h (Figure 1). Therefore, the samples with different groups at 18 h and 28 h were compared, respectively, to identify the potential functional SMB. Eight prokaryotes at the family level, including *Flavobacteriaceae*, *Enterococcaceae*, *Caulobacteraceae*, *Rhodobacteraceae*, *Campylobacteraceae*, *Sphingomonadaceae*, *Cryomorphaceae*, and *Burkholderiaceae*, showed significant abundance differences ($p < 0.05$) at 18 h. Besides *Flavobacteriaceae*, *Enterococcaceae*, *Caulobacteraceae*, *Rhodobacteraceae*, *Sphingomonadaceae*, and *Burkholderiaceae*, four prokaryotes (family level), including *Comamonadaceae*, *Bacteriovoracaceae*, *Alcaligenaceae*, and *Methylophilaceae*, also showed significant differences in abundance ($p < 0.05$) at 28 h. In addition, there were other non-SMB bacteria (*Moraxellaceae*, *Sporichthyaceae*, and *Sphingobacteriaceae*) that differed significantly over time in the control and experimental groups (Figure 3). Table 2 summarizes the relative microorganisms associated with sulfur metabolism in the reported literature. *Flavobacteriaceae*, *Enterococcaceae*, *Sphingomonadaceae* and *Cryomorphaceae* are SRB;

*Bacteriovoracaceae, Rhodobacteraceae, Campylobacteraceae, Comamonadaceae, Alcaligenaceae,* and *Methylophilaceae are* SOB; *Caulobacteraceae* and *Burkholderiaceae* belong to SRB and SOB.

**Table 2.** SMB associated with sulfur metabolism from the literature.

| Role | SMB | Main Substrate Utilization | Protein Description | Gene | References/Sources |
|---|---|---|---|---|---|
| SRB | *Flavobacteriaceae* | $SO_4^{2-}$ | Thioredoxin reductase | *trh_1* and *trxB* | [30] https://www.ncbi.nlm.nih.gov/proteinclusters (accessed on 28 January 2017) |
| | *Enterococcaceae* | $SO_4^{2-}$ | Sulfur reduction protein DsrE | *EFPG_01082, EFUG_01619, HMPREF1348_01292, EFWG_02410, EfmE1039_2332, W75_03680* and etc. | [31] https://www.ncbi.nlm.nih.gov/proteinclusters (accessed on 28 January 2017) |
| | *Sphingomonadaceae* | $SO_4^{2-}$ | Thioredoxin reductase | *cysI* | [32] |
| | *Cryomorphaceae* | $SO_4^{2-}$ | Thioredoxin reductase | *trh_1* and *trxB* | [30] https://www.ncbi.nlm.nih.gov/proteinclusters (accessed on 29 January 2017) |
| SOB | *Bacteriovoracaceae* | $S_2O_3^{2-}$ | Sulfur transport, 4Fe-4S dicluster domain protein, and NADH: ubiquinone oxidoreductase | Unknown | https://www.ncbi.nlm.nih.gov/proteinclusters (accessed on 28 January 2017) |
| | *Rhodobacteraceae* | $S^{2-}, S^0, S_2O_3^{2-}$ and $SO_3^{2-}$ | Sulfur oxidation protein SoxX | *A33M_4449* | [15,30,32,33] |
| | *Campylobacteraceae* | $S^{2-}, S^0$ and $S_2O_3^{2-}$ | Sox and Sqr | Unknown | [19,34] |
| | *Comamonadaceae* | $S^{2-}$ | Sulfur oxidation protein SoxY | *CTATCC11996_02647, CtCNB1_3856, CtesDRAFT_PD0732, COMTE_24065* and *CTS44_11486* | [8,33] |
| | *Alcaligenaceae* | $S^{2-}, S^0$ and $S_2O_3^{2-}$ | Sulfur oxidation protein SoxY | *QWA_06910* and *C660_15968* | [19,33] |
| | *Methylophilaceae* | $S^{2-}$ | SoxZ and sulfite reductase | *cysI, nir* and *sir* | https://www.ncbi.nlm.nih.gov/proteinclusters (accessed on 19 February 2017) |
| SRB or SOB | *Caulobacteraceae* | SRB: $SO_3^{2-}$ / SOB: $S^{2-}$ | Sulfite reductase/ Unknown | *cysI, cysI1, cysI2, cysI_1, cysI_2, nirA, sir, sir1, sir11* and *sir2*/ Unknown | https://www.ncbi.nlm.nih.gov/proteinclusters (accessed on 19 February 2017) |
| | *Burkholderiaceae* | SRB: $SO_4^{2-}$ / SOB: $S^{2-}$ and $S^0$ | Unknown/ SoxY and SoxZ | Unknown/ *B025_06765, BRPE64_BCDS01700, BURK_036384, BYI23_B014570* and *BurJ1DRAFT_3069* | [34]/ [15,32] |

### 3.4. Identification of SOB and Their Performance

The Pearson correlation analysis between the above series of sulfides and SMB was shown in Table 3. Nine prokaryotes showed significant correlations with some sulfide species in the present study, and a positive correlation ($r = 0.900$, $p < 0.05$) was found between *Enterococcaceae* and $S^{2-}$. Meanwhile, *Enterococcaceae* showed a strong negative correlation with both $S^0$ ($r = -0.969$, $p < 0.01$) and $SO_4^{2-}$ ($r = -0.977$, $p < 0.01$). *Comamonadaceae, Burkholderiaceae, Alcaligenaceae, Methylophilaceae,* and *Caulobacteraceae* showed significant correlations with both $S^{2-}$ and $SO_4^{2-}$, respectively; their correlation indices indicated that the $S^{2-}$ oxidation and $SO_4^{2-}$ accumulation mainly depended on these microorganisms. Furthermore, *Comamonadaceae, Alcaligenaceae, Methylophilaceae,* and *Caulobacteraceae* also showed strong positive correlations to $S^0$, which could be inferred that these four prokaryotes played important roles in the production of $S^0$ as well. In addition, there was a notable correlation between *Campylobacteraceae* and $S_2O_3^{2-}$ ($r = 0.809$, $p < 0.05$), demonstrating that it was attributed to $S_2O_3^{2-}$ generation via $S_4$ Intermediate ($S_4$I) pathway [13]. While *Bacteriovoracaceae* and *Rhodobacteraceae* showed a significant correlation with $SO_3^{2-}$, suggesting that *Bacteriovoracaceae* and *Rhodobacteraceae* were associated with $SO_3^{2-}$ accumulation.

**Table 3.** Pearson correlations between series of sulfides and SMB (family level).

| | $S^{2-}$ | $S^0$ | $S_2O_3^{2-}$ | $SO_3^{2-}$ | $SO_4^{2-}$ | Sphingomonadaceae | Rhodobacteraceae | Flavobacteriaceae | Enterococcaceae | Cryomorphaceae | Comamonadaceae | Campylobacteraceae | Burkholderiaceae | Alcaligenaceae | Methylophilaceae | Caulobacteraceae | Bacteriovoracaceae |
|---|---|---|---|---|---|---|---|---|---|---|---|---|---|---|---|---|---|
| $S^{2-}$ | 1 | | | | | | | | | | | | | | | | |
| $S^0$ | −0.965 ** | 1 | | | | | | | | | | | | | | | |
| $S_2O_3^{2-}$ | 0.102 | −0.038 | 1 | | | | | | | | | | | | | | |
| $SO_3^{2-}$ | −0.508 | 0.483 | −0.062 | 1 | | | | | | | | | | | | | |
| $SO_4^{2-}$ | −0.965 ** | 0.999 ** | −0.073 | 0.508 | 1 | | | | | | | | | | | | |
| Sphingomonadaceae | −0.737 | 0.782 | −0.585 | 0.212 | 0.797 | 1 | | | | | | | | | | | |
| Rhodobacteraceae | −0.776 | 0.705 | −0.414 | 0.852* | 0.731 | 0.611 | 1 | | | | | | | | | | |
| Flavobacteriaceae, | 0.320 | −0.438 | 0.558 | −0.554 | −0.474 | −0.676 | −0.612 | 1 | | | | | | | | | |
| Enterococcaceae | 0.900 * | −0.969 ** | 0.177 | −0.538 | −0.977 ** | −0.854 * | −0.744 | 0.645 | 1 | | | | | | | | |
| Cryomorphaceae | −0.352 | 0.233 | 0.653 | 0.393 | 0.212 | −0.373 | 0.220 | 0.510 | −0.042 | 1 | | | | | | | |
| Comamonadaceae | −0.906 * | 0.925 * | 0.305 | 0.374 | 0.908 * | 0.521 | 0.511 | −0.101 | −0.814 * | 0.514 | 1 | | | | | | |
| Campylobacteraceae | 0.664 | −0.606 | 0.809 * | −0.307 | −0.631 | −0.894 * | −0.740 | 0.609 | 0.673 | 0.314 | −0.312 | 1 | | | | | |
| Burkholderiaceae | −0.833 * | 0.925 | −0.270 | 0.395 | 0.934 * | 0.922 * | 0.650 | −0.685 | −0.979 ** | −0.143 | 0.736 | −0.712 | 1 | | | | |
| Alcaligenaceae | −0.813 * | 0.856 * | −0.538 | 0.488 | 0.866 * | 0.957 ** | 0.800 | −0.765 | −0.925 * | −0.212 | 0.581 | −0.890 | 0.942 ** | 1 | | | |
| Methylophilaceae | −0.902 * | 0.963 ** | −0.139 | 0.631 | 0.972 ** | 0.800 | 0.792 | −0.649 | −0.933 ** | 0.117 | 0.817 * | −0.639 | 0.951 ** | 0.904 * | 1 | | |
| Caulobacteraceae | −0.850 * | 0.880* | −0.494 | 0.451 | 0.897 * | 0.964 ** | 0.781 | −0.710 | −0.940 ** | −0.170 | 0.638 | −0.881 * | 0.955 ** | 0.996 ** | 0.915 * | 1 | |
| Bacteriovoracaceae | −0.562 | 0.501 | −0.133 | 0.988 ** | 0.526 | 0.247 | 0.901 * | −0.508 | −0.538 | 0.423 | 0.389 | −0.389 | 0.392 | 0.516 | 0.628 | 0.414 | 1 |

* Correlation is significant at the 0.05 level (1-tailed); ** Correlation is significant at the 0.01 level (1-tailed).

Compared with other microorganisms, *Enterococcaceae* has the highest positive correlation with $S^{2-}$. The relationship between these SMBs was also observed in Table 3. A competitive relationship was found between *Enterococcaceae* and five SOB, including *Comamonadaceae*, *Burkholderiaceae*, *Alcaligenaceae*, *Methylophilaceae*, and *Caulobacteraceae*, based on significantly negative correlations. Moreover, the correlation indices of these four SOBs showed a significant relationship between *Burkholderiaceae*, *Alcaligenaceae*, *Methylophilaceae*, and *Caulobacteraceae*, while *Comamonadaceae* only had a notable relationship with *Methylophilaceae*. Meanwhile, there was a significant relationship between *Bacteriovoracaceae* and *Rhodobacteraceae*. The bio-oxidation of $S^{2-}$ is the result of the combined action of the SMBs, but the contribution of different microorganisms varies significantly. As discussed above, it was prudent to conclude that *Comamonadaceae*, *Burkholderiaceae*, *Alcaligenaceae*, *Methylophilaceae*, *Caulobacteraceae*, *Campylobacteraceae*, *Bacteriovoracaceae*, and *Rhodobacteraceae* were the main SOB involved in this inorganic sulfur oxidation processes (Figure 4a).

The detailed $S^{2-}$ oxidation process and the participation of the SOB could be summarized in Figure 4b. Figure 4b is based on the microbial diversity and the microorganisms' differences along with the abundance difference and the sulfide changes through Pearson correlation analysis to establish correlations between the microorganisms and the sulfur oxidation process and to speculate on possible pathways and possible key microorganisms for each pathway. The percentages marked in this figure represent the percentage ratio of the changes in inorganic sulfides' concentration at this instant compared to the previous instant, and the microorganisms next to inorganic sulfides indicate that they might play a major role in that inorganic sulfide change. Throughout the experiment, the $S^{2-}$ was oxidized to other inorganic sulfates mainly through the effect of *Comamonadaceae*, *Burkholderiaceae*, *Alcaligenaceae*, *Methylophilaceae*, and *Caulobacteraceae*, resulting in a continuous decrease in $S^{2-}$ concentration in the system. In the first 18 h, The concentration of $S^0$ accumulated rapidly, which was mainly due to the action of *Comamonadaceae*, *Alcaligenaceae*, *Methylophilaceae*, and *Caulobacteraceae*. In particular, the average generation rate of $S^0$ reached 100% per hour during 6–18 h, which was significantly higher than other inorganic sulfur concentration changes. However, after 18 h, $S^0$ was oxidized to other inorganic sulfides by *Campylobacteraceae*, *Bacteriovoracaceae*, and *Rhodobacteraceae*. At the same time, the concentration of $S_2O_3^{2-}$ accumulated rapidly under the effect of *Campylobacteraceae*; but the consumption of $S_2O_3^{2-}$ in the last 28 h was higher than the consumption of other inorganic sulfides, which was mainly due to the action of *Burkholderiaceae*, *Methylophilaceae*, and *Bacteriovoracaceae*. The production of $SO_3^{2-}$ occurs mainly during 6–28 h, which was associated with *Bacteriovoracaceae* and *Rhodobacteraceae*, and the generation of $SO_3^{2-}$ was the dominant sulfides oxidization process during 18–28 h. In contrast to the change in $S^{2-}$ concentration, the concentration of $SO_4^{2-}$ kept increasing throughout the experiment, this was promoted by *Comamonadaceae*, *Burkholderiaceae*, *Alcaligenaceae*, *Methylophilaceae*, and *Caulobacteraceae*; the generation of $SO_4^{2-}$ was the main sulfide oxidization process in the first 6 h.

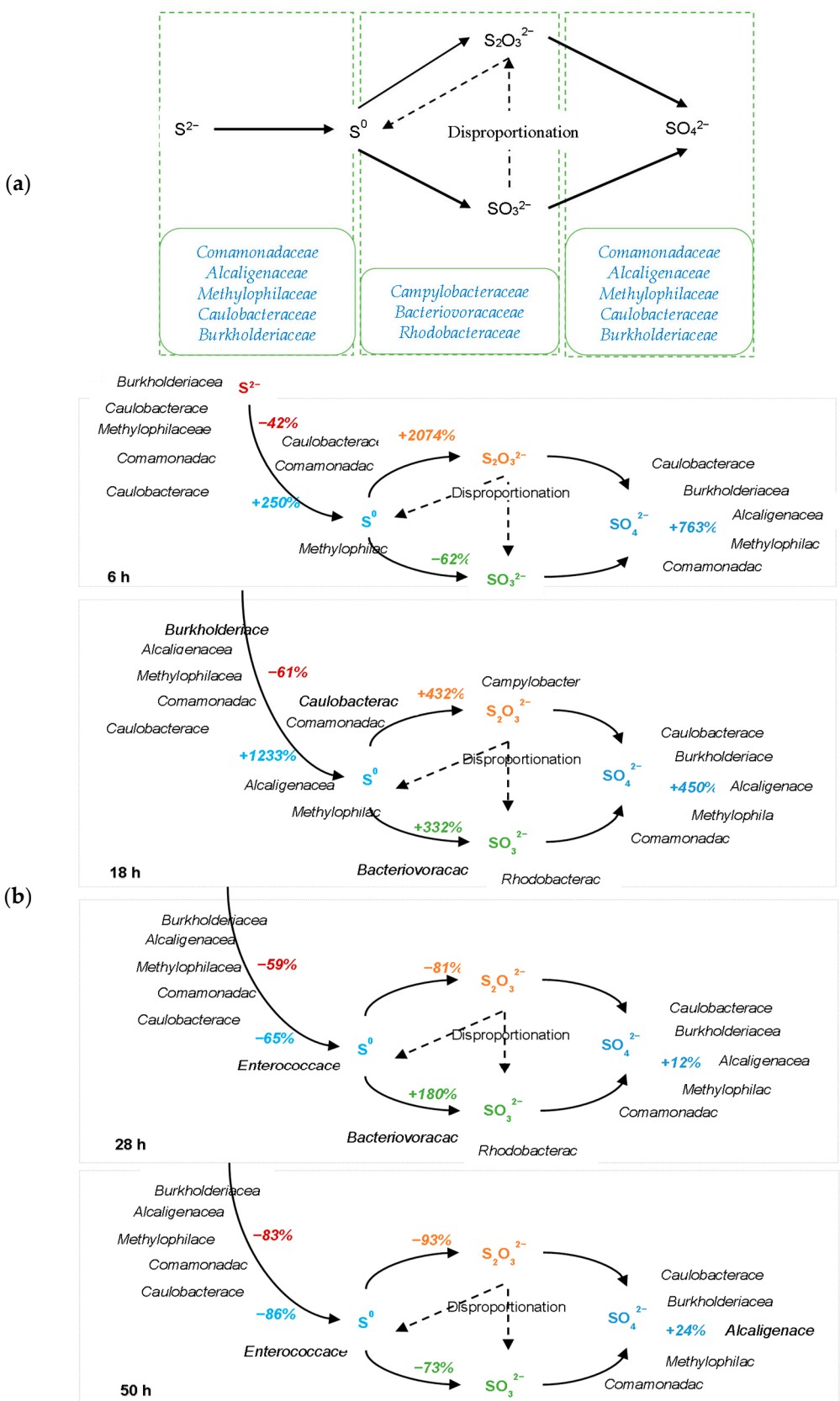

**Figure 4.** Inorganic sulfur-oxidizing process (**a**) inorganic sulfur-oxidizing pathway and involving SOB; (**b**) performance of SOB and sulfates' oxidizing process during each period.

## 4. Discussion

The results of variation of inorganic sulfur indicated that unstable lower-valence sulfur ion species including $S^{2-}$, $S^0$, $S_2O_3^{2-}$, and $SO_3^{2-}$ were consumed to generate stable higher-valence $SO_4^{2-}$, and sulfur-oxidizing became the main metabolic path of inorganic sulfur bio-conversion. Moreover, both sulfur oxidation and sulfur reduction processes were present simultaneously in the bio-treatment group and the control group, which led to different forms of sulfur ions being detected during the reaction. In the control group, the concentration of $SO_3^{2-}$ increased in the first 8 h and the concentration of $S^0$ increased in the first 20 h, both of them remained stable in the subsequent reaction stages; the concentration of $S^{2-}$ first decreased then remained stable after 30 h; the change in the concentration of $SO_4^{2-}$ and $S_2O_3^{2-}$ was negligible during the reaction. This indicated that although the sulfur oxidation process was produced in the control group at the early stage of the reaction, it was not obvious compared with the bio-treatment group. It could be concluded that the inorganic sulfur-oxidizing process was efficiently augmented by MCMC.

From Table 1, the Ace indices in both groups showed an increase followed by a decrease. However, from 28 h, the Ace indices in the group with the MCMC were higher than the control group, which might be due to the activation of some indigenous microorganisms by sulfur-oxidizing composite microorganisms so that these microorganisms could use the substrate and proliferate rapidly. The Shannon indices in both groups showed similar trends, except at 18 h, indicating that the biodiversity was similar most of the time, but at 18 h the bio-treatment group was higher than that in the control group. The difference at 18 h was probably due to the gradual enrichment of a few inorganic SOB in the system, and thus the removal of $S^{2-}$ might be achieved mainly by the action of these microorganisms [35,36]. The Coverage index reached above 0.997 in both groups, indicating good sequence detection in the samples. In addition, the PCoa in Figure 2 demonstrated that the bacterial communities were most affected by the mixed culture stimulation, with 53.99% of PC1 at the phylum level. That is, the MCMC did not become the dominant microorganism, it just simulated the structure change of the original microorganisms to enhance the sulfur oxidation process. From an ecological point of view, higher microbial community diversity can be considered an augmentation of functional redundancy [37]. Microbial communities with greater evenness are likely to be more functional, which is a key factor in maintaining the functional stability of ecosystems [37]. However, in this study, the new microbial community was shaped after inoculating the MCMC, some of the abundance of the original SRBs was reduced and some of the original SOBs were activated. Furthermore, the bio-augmentation group was generally less diverse and homogeneous than the control group, which seemed to activate the key for inorganic sulfur oxidation and enabled the microbes to cope with the substrate change with the Gibbs free energy changes during the reaction [38,39]. This feature facilitated the optimization of bacterial communities for inorganic sulfur oxidation [40].

From the analysis of sulfide metabolism bacteria, the differences in microbial abundance between the two groups could imply that the MCMC in the bio-treatment group activated the potential sulfur-oxidizing microorganisms. The analysis of sulfide metabolism bacteria revealed that *Enterobacteriaceae*, *Brucellaceae*, and *Xanthomonadaceae* also showed significant differences in abundance between the two groups as the response progressed (mean sums outside the top 10 species). The eight microorganisms (*Flavobacteriaceae*, *Enterococcaceae*, *Caulobacteraceae*, *Rhodobacteraceae*, *Campylobacteraceae*, *Sphingomonadaceae*, *Cryomorphaceae*, and *Burkholderiaceae*) have demonstrated the ability to metabolize inorganic sulfur (Table 2). For instance, *Flavobacteriaceae*, *Enterococcaceae*, *Sphingomonadaceae*, *Cryomorphaceae*, and *Caulobacteraceae* were often thought to be associated with SRB because of the presence of thioredoxin reductase, sulfur reduction protein DsrE or sulfite reductase [30,32], and they could use $SO_4^{2-}$ as the substrate as well. Furthermore, *Bacteriovoracaceae*, *Rhodobacteraceae*, *Campylobacteraceae*, *Burkholderiaceae*, *Comamonadaceae*, *Alcaligenaceae*, and *Methylophilaceae* usually represented SOB in other studies [19,33,41]. The reduced sulfides such as $S^{2-}$, $S^0$, and $S_2O_3^{2-}$ could be used as electron donors. Interestingly, sometimes *Burkholderiaceae* and

*Caulobacteraceae* could also play opposite roles, depending on the circumstance they are located in [34,42]. Moreover, from the literature, the non-SMB bacteria with a significant abundance change: *Moraxellaceae* and *Sphingobacteriaceae* are aerobic bacteria and *Sporichthyaceae* is facultative anaerobes bacteria, these microorganisms might play a supporting role in the sulfur oxidation process [43–45]. It could be inferred that the abundance differences between these non-SMB in the bio-treatment group and the control group were mainly due to the reproduction of these microorganisms being inhibited in the bio-treatment group, where more oxygen was consumed as the sulfur oxidation process was accelerated by the MCMC.

Moreover, the results of the identification and the performance of SOB suggest that the addition of mixed culture could accelerate the inorganic sulfur oxidization reactions by stimulating SMB. The roles played by these SMB remain unclear. The Pearson correlation analysis in Table 3 indicates that the abundance of *Enterococcaceae* tends to decrease with the removal of $S^{2-}$ and the accumulation of $SO_4^{2-}$, suggesting that *Enterococcaceae* could either grow or obtain energy by reducing $SO_4^{2-}$ as an electron acceptor [12,16]. Sulfur redox is a dynamic and reversible process, where $S^{2-}$ is eventually oxidized to $SO_4^{2-}$ by SOB, while $SO_4^{2-}$ is also reduced to $S^{2-}$ by SRB during the reaction. The production of $S^{2-}$ during the inorganic sulfur oxidation process is mainly derived from the disproportionation of $S_2O_3^{2-}$ [19]. From the Pearson analysis above, *Enterococcaceae* has the highest positive correlation with $S^{2-}$. Therefore, it is hypothesized that *Enterococcaceae* may play an important role in regulating the $S_2O_3^{2-}$ disproportionation reaction, contributing to the production of $S^{2-}$. In addition, some literature studies indicated that both *Comamonadaceae* and *Alcaligenaceae* could participate in the oxidation of $S^{2-}$ with increasing $SO_4^{2-}$ concentration [8,19,32]. However, a further study has shown that *Alcaligenaceae* was also able to oxidize $S_2O_3^{2-}$ through aerobic and anaerobic [19]. *Campylobacteraceae* was proved that could utilize $S^0$ as an electron acceptor and form polysulfide [46]. It should be noted that *Enterococcaceae*, *Burkholderiaceae*, and *Caulobacteraceae* acted as SOB during the process of this work. These results of the action of *Enterococcaceae*, *Burkholderiaceae*, and *Caulobacteraceae* were different from those described by Llorens-Marès et al. and NCBI (Table 3) [34]. This might be due to the fact that this experiment was operated under aeration, whereas other results obtained in the literature under natural conditions might create an anaerobic or anoxic environment [2,12,34]. The dissolved oxygen (DO) in water samples in this study might be higher than DO in the environment under natural conditions. The application of sufficient oxygen might facilitate the oxidative conversion of $S^{2-}$ to $SO_4^{2-}$, and the activation of the associated inorganic sulfur oxidases. Some limitations of this study were that some of the pathways of sulfide oxidation regulated by SOB were not clear. For instance, $S_2O_3^{2-}$ could be generated by $S^0$ oxidation and/or $SO_3^{2-}$ disproportionation [12,47]. It should be noted that this work only demonstrated that *Campylobacteraceae* was associated with the production of $S_2O_3^{2-}$, but there was little evidence as to which pathway was primarily or actually regulated by the *Campylobacteraceae*.

## 5. Conclusions

Considerable amounts of $S^{2-}$ are emitted from many industrial and agricultural activities. The slow inorganic sulfur-oxidizing process under natural conditions enables $S^{2-}$ to accumulate massively and induces natural water bodies pollution, even making water blacken. The common occurrence of water pollution has become a serious problem around the world. Biological desulfurization is a highly effective, economically viable alternative method, which has indeed been a hot topic in modern environmental science research. However, to date, there is little information about the fundamental research work of mixed culture for $S^{2-}$ removal in a heavily polluted river.

This study reported that mixed culture with *Citrobacter* sp.sp1, *Ochrobactrum* sp.sp2, and *Stenotrophomonas* sp.sp3 had nearly 98% of $S^{2-}$ removal efficiency within 50 h. The mechanism involved in bio-augmentation could be summarized as follows. The native microbial community dynamics were stimulated and simplified by inoculated mixed

culture to cope with the sulfides change. *Enterococcaceae*, *Comamonadaceae*, *Burkholderiaceae*, *Alcaligenaceae*, *Methylophilaceae*, *Caulobacteraceae*, *Campylobacteraceae*, *Bacteriovoracaceae*, and *Rhodobacteraceae* performed as SOB for accelerating inorganic sulfur-oxidizing progress in this experiment. *Comamonadaceae*, *Burkholderiaceae*, *Alcaligenaceae*, *Methylophilaceae*, and *Caulobacteraceae* played important roles in $S^{2-}$ oxidization and $SO_4^{2-}$ accumulation across the whole process. Except for *Burkholderiaceae*, the other four SOB also made contributions to $S^0$ generation in the first 18 h, but *Enterococcaceae* was in charge of the oxidation of $S^0$ to other sulfides after 18 h. *Campylobacteraceae* took effect on $S_2O_3^{2-}$ accumulation from 6 h to 18 h. *Bacteriovoracaceae* and *Rhodobacteraceae* took effect when $SO_3^{2-}$ increased from 6 h to 28 h. This study made a high-efficiency microbial consortium to treat the black-odor water bodies and found that the microbial consortium could enhance the sulfur oxidation process by changing the structure of the original microorganisms in the black-odor water body. This study also demonstrated that the mixed culture microbial consortium might have the potential to be used to remove $S^{2-}$ and clarify water in the actual black-odor water bodies and the wastewaters with high concentrations of $S^{2-}$.

**Author Contributions:** Conceptualization, C.S. and X.L.; methodology, C.S. and X.L.; software, C.S. and Y.S.; validation, Y.S., H.G. and X.L.; formal analysis, C.S.; investigation, C.S.; resources, C.S.; data curation, C.S.; writing—original draft preparation, C.S.; writing—review and editing, Y.S.; visualization, Y.S.; supervision, X.L. and P.L.; project administration, X.L.; funding acquisition, X.L. All authors have read and agreed to the published version of the manuscript.

**Funding:** This research was funded by the Natural Science Foundation of Beijing, China, grant number 8182058 and the Central Level, Scientific Research Institutes for Basic R&D Special Fund Business, grant number 2019YSKY003.

**Institutional Review Board Statement:** Not applicable.

**Informed Consent Statement:** Not applicable.

**Data Availability Statement:** The data presented in this study are available in this article.

**Conflicts of Interest:** The authors declare no conflict of interest.

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
