# Peer review of "Bio-Augmentation of S2− Oxidation for a Heavily Polluted River by a Mixed Culture Microbial Consortium"

_fermentation, doi:10.3390/fermentation9070592_

Round 1
Reviewer 1 Report
This paper, entitled “Bio-augmentation of S2- oxidation for a heavily polluted river by a mixed culture microbial consortium”, is a scholarly work. The content is relevant to Fermentation Journal. The paper addresses an interesting topic and can be published without minor corrections. The reviewer single suggestions is to add the references in page 2 lines 74 and 92.
Author Response
Dear Reviewer,
Thank you for the valuable comments on our manuscript entitled “Bio-augmentation of S2- oxidation for a heavily polluted river by a mixed culture microbial consortium” (Reference No: fermentation-2387366). We also very appreciate for your recognition and encouragement of our manuscript. We have given a point-by-point list of our response to the comment below. We have also revised the references, we used the “Track Changes” function in MS Word and all the revisions in the revised manuscript have been marked in red color. The revised manuscript is in the attachment.
Best regards,
Chen Song, Yajun Shi, Hongjie Gao, Ping Liu and Xiaoling Liu
Response to Reviewer 1 Comments
Point 1: The reviewer single suggestions is to add the references in page 2 lines 74 and 92.
Response 1: Done. Thanks again for your encouragement. The references are added in the revised manuscript, respectively (Line 94, Page 2; Line 113, Page 3).

Reviewer 2 Report
Abstract
1. Line 15, page 1 – the authors may wish to explain what ‘used to active sulfate oxidizing bacterium’ means. Does it mean to activate? The authors may provide a summary of how the activation was achieved.
2. Line 18, page 1 – ‘sulfides’ species’ should be ‘sulfide species’
3. Line 23, page 1 – does ‘abundance difference’ mean the ‘difference in microbial abundance or sulfides abundance’?
4. Line 24, page 1 – the plural form of ‘function’, i.e. ‘functions’ is more appropriate.
Introduction
1. Line 32, page 1 – ‘intensification’ is a more appropriate word to replace ‘strengthening’.
2. The authors may explain if the levels of sulfide (3.15 mg/L to 17.93 mg/L) were considered high with reference to a certain standard.
3. The authors may further justify why accumulation of S2- is a pressing problem in rivers besides its impact of the color of river water.
4. Line 64, page 2 – ‘has been proved’ should be ‘have been proven’.
5. The authors may perform a brief literature review and summarize the work done in isolation of SOBs to highlight the research gap that justifies this study.
6. The authors may explain the merits of sourcing SOB from a polluted river instead of other sources to highlight the importance of this study.
7. Line 72, page 2 - The authors may explain the interest in looking at the three SOBs mentioned instead of other SOBs and whether the prevailing SOBs are different for different rivers.
Methods
1. The authors may explain why the water from Dongsha River was sampled, what were the water quality parameters analyzed and why.
2. Line 111, page 3 – ‘inoculated 0.1 g/L of’ should be ‘inoculated with 0.1 g/L of’
3. The authors may specify what the Chinese Standards for analyzing different sulfur species are. Were other detection methods with higher sensitivity, e.g. ICP-AES, AAS considered?
Results
1. Line 168, page 4 – Revise the sentence starting with ‘Similarly, both of….’ as it seems syntactically flawed.
2. Figure (b) – The authors may explain why the formation of S and SO32- was significant in the control despite the lower abundance of SOBs and the higher abundance of certain SRBs.
3. Since measuring was repeated in triplicate, the authors may report the concentrations as mean ± standard deviation.
4. Line 182, page 5 – the authors mention two processes for inoculation with special strains. Did the processes result in different community structures, hence different concentrations of different sulfur species examined? Were the two processes investigated separately in the study?
5. Line 214, page 6 – revise the sentence starting with ‘While the abundance of ….’ as it appears syntactically flawed.
6. There are numerous spelling errors throughout the text, e.g. Line 245, page 7 – ‘summrises’ should be ‘summarises’; Line 289, page 10 – sulfieds’ change should be sulfide change etc. Kindly proofread the text again.
7. Line 276, page 9 – The authors may explain why correlation indices could point to synergistic effects since correlation does not indicate causation. It only shows the existence of a relationship.
8. Line 290, page 9 – If S2- were changed to other inorganic sulfides, there would be neither oxidation nor reduction? Perhaps inorganic sulfides should be inorganic sulfates or inorganic sulfur species.
9. Line 304, page 10 – the sentence containing ‘which contributed to….’ lacks clarity. Kindly revise.
10. The authors may explain how Figure 4(b) was constructed and whether the inorganic sulfur oxidizing pathway is preliminary or confirmed since only a correlational analysis was conducted to connect different sulfur species to different bacteria.
Discussion
1. Line 339, page 12 – the authors may explain what ‘functional groups of inorganic sulfur metabolizing bacteria’ means since functional groups are usually associated with chemical compounds.
2. Line 360, page 12 – check and revise the sentence.
3. The authors may consider comparing the findings of this study with more studies. Only one study has been compared to in this section.
Conclusion
1. Sulfur contamination is only one of the many types of water pollution. The authors may explain why desulfurization is important to mitigate water pollution as a whole.
2. The authors may highlight the novelty and practical implications of this study.
The authors may wish to perform a thorough proofreading to remove the obvious grammatical, syntactical and spelling errors.
Author Response
Dear Reviewer,
Thank you for the valuable comments on our manuscript entitled “Bio-augmentation of S2- oxidation for a heavily polluted river by a mixed culture microbial consortium” (Reference No: fermentation-2387366). We also appreciate for your detailed suggestions on our manuscript, these comments are very valuable to us. We have given a point-by-point list of our response to the comments in the attachment. We have also revised all the errors according to the comments very carefully, we used the “Track Changes” function in MS Word and all the revisions in the revised manuscript have been marked in red color. The revised manuscript is in the attachment.
Thanks again for the valuable comments,
Best regards,
Chen Song, Yajun Shi, Hongjie Gao, Ping Liu and Xiaoling Liu

Reviewer 3 Report
The study by Song et al. addresses aquatic pollution with high sulfide concentrations, which is a toxic compound. The authors investigated the effect of the addition of a consortium of bacteria (Citrobacter sp., Ochrobactrum sp. and Stenotrophomonas sp.) to water samples from the Dongsha River, which has a high sulfide concentration. The did chemical analyses of inorganic sulfur species and 16S amplicon sequencing as community analysis. Based on the 16S profiles, they provide predictions which microorganism may be involved in transformation of inorganic sulfur and by which pathway.
Critique: While the authors addressed an important environmental problem, there are critical issues with the overall aim and there are numerous points in the presentation that should be improved substantially. First, it is unclear how an application of the consortium in the field would be successful given that some sort of aeration would be needed. Then, it is not clear what the in situ function of the consortium actually is. The provided inoculum size is not helpful (how many cells were in 0.1 g/L?), it is not clear whether there was any carry-over of consortium medium, and the strains of the consortium were not quantified during the course of the experiment. Moreover, if we would assume that the members of the consortium did indeed oxidized sulfide in the river sample, then it would have been an experiment with a rather predictable outcome: removal of toxic sulfide triggered changes in the overall community. The authors’ prediction on which phylotype might catalyze which sulfur transformation needs further evidence. All respective families contain members that are unable to carry out catabolic sulfur transformation, and it is not known which particular members of a family were present in the experimental set-up. Furthermore, the presence of few or even only one homolog in a microbe to a protein involved in inorganic sulfur transformation does not necessarily mean that this microbe can actually carry out the transformation (i.e. all proteins of the respective pathway are needed). Moreover, some predictions are questionable, e.g.: Table 2, Flavobacteriaceae, thioredoxin reductase involved dissimilatory sulfate reduction??).
Additional comments
Line 4: sulfate-reducing bacteria
L53: thiosulfate and sulfur are not products of biological sulfate reduction
L74: Please cite your previous work
Fig 1: Please convert “mg/L” into “mg/L of S” or molarity to ease comparison of concentrations
Extensive editing is needed.
Author Response
Dear Reviewer,
Thank you for the valuable comments on our manuscript entitled “Bio-augmentation of S2- oxidation for a heavily polluted river by a mixed culture microbial consortium” (Reference No: fermentation-2387366). We have given a point-by-point list of our response to the comments and the questions in critique in the attachment. We have also revised all the errors according to the comments very carefully, we used the “Track Changes” function in MS Word and all the revisions in the revised manuscript have been marked in red color. The revised manuscript is in the attachment.
With the best wishes,
Chen Song, Yajun Shi, Hongjie Gao, Ping Liu and Xiaoling Liu
